# VilNMN: A Neural Module Network approach to Video-Grounded Language Tasks

## Abstract

Neural module networks (NMN) have achieved success in image-grounded tasks such as Visual Question Answering (VQA) on synthetic images. However, very limited work on NMN has been studied in the video-grounded language tasks. These tasks extend the complexity of traditional visual tasks with the additional visual temporal variance. Motivated by recent NMN approaches on image-grounded tasks, we introduce Visio-Linguistic Neural Module Network (VilNMN) to model the information retrieval process in video-grounded language tasks as a pipeline of neural modules. VilNMN first decomposes all language components to explicitly resolve any entity references and detect corresponding action-based inputs from the question. The detected entities and actions are used as parameters to instantiate neural module networks and extract visual cues from the video. Our experiments show that VilNMN can achieve promising performance on two video-grounded language tasks: video QA and video-grounded dialogues.

## 1 Introduction

Vision-language tasks have been studied to build intelligent systems that can perceive information from multiple modalities, such as images, videos, and text. Extended from imaged-grounded tasks, e.g. (Antol et al., 2015), recently Jang et al. (2017); Lei et al. (2018) propose to use video as the grounding features. This modification poses a significant challenge to previous image-based models with the additional temporal variance through video frames. Recently Alamri et al. (2019) further develop video-grounded language research into the dialogue domain. In the proposed task, *video-grounded dialogues*, the dialogue agent is required to answer questions about a video over multiple dialogue turns. Using Figure 1 as an example, to answer questions correctly, a dialogue agent has to resolve references in dialogue context, e.g. "he" and "it", and identify the original entity, e.g. "a boy" and "a backpack". In addition, the dialogue agent also needs to identify the actions of these entities, e.g. "carrying a backpack" to retrieve information along the temporal dimension of the video.

Current state-of-the-art approaches to video-grounded language tasks, e.g. (Le et al., 2019b; Fan et al., 2019) have achieved remarkable performance through the use of deep neural networks to retrieve grounding video signals based on language inputs. However, these approaches often assume the reasoning structure, including resolving references of entities and detecting the corresponding actions to retrieve visual cues, is implicitly learned. An explicit reasoning structure becomes more beneficial as the tasks complicates in two scenarios: video with complex spatial and temporal dynamics, and language inputs with sophisticated semantic dependencies, e.g. questions positioned in a dialogue context. In these cases, it becomes challenging to interpret model outputs, assess model reasoning capability, and identify errors in neural network models.

Similar challenges have been observed in image-grounded tasks in which deep neural networks often exhibit shallow understanding capability as they simply exploit superficial visual cues (Agrawal et al., 2016; Goyal et al., 2017; Feng et al., 2018; Serrano & Smith, 2019). Andreas et al. (2016b) propose neural model networks (NMNs) by decomposing a question into sub-sequences called *program* and assembling a network of neural operations. Motivated by this line of research, we propose an NMN approach to video-grounded language tasks. Our approach benefits from integrating neural networks with a compositional reasoning structure to exploit low-level information signals in video. An example of the reasoning structure can be seen on the right side of Figure 1.

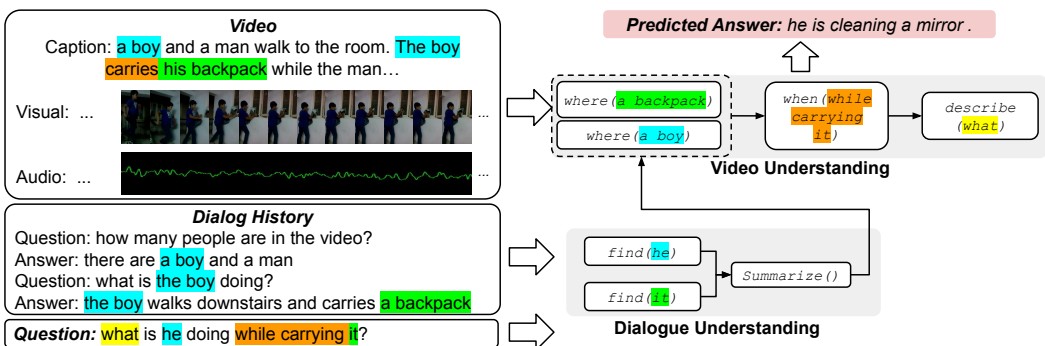

Figure 1: A sample video-grounded dialogue: Inputs are question, dialogue history, video with caption, visual and audio input, and the output is the answer to the question. On the right side, we demonstrate an example symbolic reasoning process a dialogue agent can perform to extract textual and visual clues for the answer.

We propose Visio-Linguistic Neural Module Network (VilNMN) for video-grounded language tasks. VilNMN leverages entity-based dialogue representations as inputs to neural operations on spatial and temporal-level visual features. Previous approaches exploit question-level and token-level representations to extract question-dependent information from video (Jang et al., 2017; Fan et al., 2019; Le et al., 2019b). In complex videos with many entities or actions, these approaches might not be optimal to locate the right features. To exploit object-level features, VilNMN is trained to identify relevant entities first, and then to extract the temporal steps using detected actions of these entities.

VilNMN is also trained to resolve any co-references in language inputs, e.g. questions in a dialogue context, to identify the original entities. Previous approaches to video-grounded dialogues often obtain question global representations in relation to dialogue context. These approaches might be suitable to represent general semantics in open-domain or chit-chat dialogues (Serban et al., 2016; Li et al., 2016) but they are not ideal to detect fine-grained entity-based information as the dialogue context evolves over time.

In summary, we introduce a neural module network approach to video-grounded language tasks through a reasoning pipeline with entity and action representations applied on the spatio-temporal dynamics of video. To cater to complex semantic inputs in language inputs, e.g. dialogues, our approach also allows models to resolve entity references to incorporate question representations with fine-grained entity information. In our evaluation, we achieve competitive performance on the large-scale benchmark Audio-visual Scene-aware Dialogues (AVSD) (Alamri et al., 2019). We also adapt VilNMN for video QA and obtain the state-of-the-art on the TGIF-QA benchmark (Jang et al., 2017) across all tasks. Our experiments and ablation analysis indicate a potential direction to develop compositional and interpretable neural models for video-grounded language tasks.

## 2 RELATED WORK

Video QA has been a proxy for evaluating a model's understanding capability of language and video and the task is treated as a visual information retrieval task. Jang et al. (2017); Gao et al. (2018); Jiang et al. (2020) propose to learn attention guided by question global representation to retrieve spatial-level and temporal-level visual features. Li et al. (2019); Fan et al. (2019); Jiang & Han (2020) model interaction between all pairs of question token-level representations and temporal-level features of input video. Extended from video QA, video-grounded dialogue is an emerging task that combines dialogue response generation and video-language understanding research. Nguyen et al. (2018); Hori et al. (2019); Hori et al. (2019); Sanabria et al. (2019); Le et al. (2019a;b) extend traditional QA models by adding dialogue history neural encoders. Kumar et al. (2019) enhances dialogue features with topic-level representations to express the general topic in each dialogue. Sanabria et al. (2019) considers the task as a video summary task and concatenates question and dialogue history into a single sequence and proposes to transfer parameter weights from a large-scale video summary model. Different from prior work, we dissect the question sequence and explicitly detect and decode any entities and their references. Our models also benefit from the additional insights on how models learn to use component linguistic inputs for extraction of visual information.

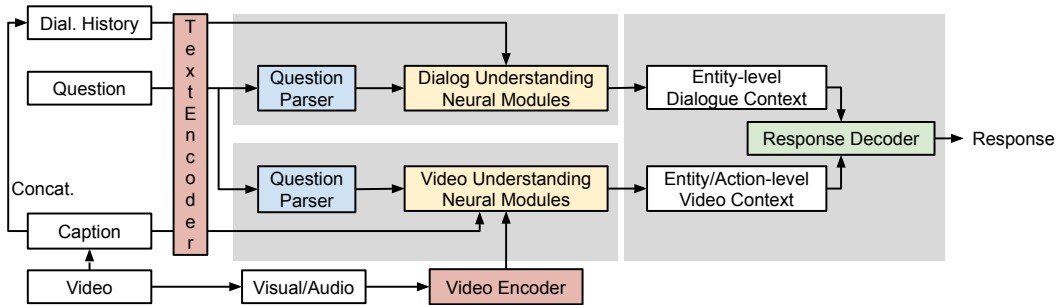

Figure 2: VilNMN includes 4 major components: (1) encoders that encode dialogue and video components into continuous vector representations; (2) question parsers that parse question of the current dialogue turn into compositional programs for dialogue and video understanding; (3) an inventory of neural modules that operate on dialogue and video input components; and (4) a response decoder that generates natural language sequence using dialogue-based and video-based execution outputs.

Extending from the line of research on neural semantic parsing (Jia & Liang, 2016; Liang et al., 2017), Andreas et al. (2016b;a) introduce NMNs to address visual QA by decomposing questions into linguistic sub-structures, known as programs, to instantiate a network of neural modules. NMN models have achieved significant success in synthetic image domains where complex reasoning process is required (Johnson et al., 2017b; Hu et al., 2018; Han et al., 2019). Our work is related to the recent work that extends NMN models to real data domains. For instance, Kottur et al. (2018); Jiang & Bansal (2019); Gupta et al. (2020) extend NMNs to visual dialogues and reading comprehension tasks. In this paper, we introduce a new approach that exploits NMN to learn dependencies between the lexical composition in language inputs and the spatio-temporal dynamics in videos. This is not present in prior NMN models which are designed to apply on a two-dimensional image input without temporal variance. In video represented as sequence of images, each represented by object-level features, applying prior NMN models require aggregating frame-level features, e.g. through average pooling, resulting in potential loss of information. An alternative solution is a late-fusion method in which an NMN model performs reasoning structure programs on sampled video frames only. An object tracking mechanism or attention mechanism is then used to fuse the output representations. Instead, we propose to construct a reasoning structure with multi-step interaction between the space-time information in video with entity-action detected in text.

## 3 METHOD

In this section, we present the design of our model, called Visio-Linguistic Neural Module Networks (VilNMN). An overview of the model can be seen in Figure 2. The input to the model consists of a dialogue $\mathcal{D}$ which is grounded on a video $\mathcal{V}$. The input components include the question of current dialogue turn $\mathcal{Q}$, dialogue history $\mathcal{H}$, and the features of input video, including visual and audio input. The output is a dialogue response, denoted as $\mathcal{R}$. Each text input component is a sequence of words $w_1, ..., w_m \in \mathbb{V}^{in}$, the input vocabulary. Similarly, the output response $\mathcal{R}$ is a sequence of tokens $w_1, ..., w_n \in \mathbb{V}^{out}$, the output vocabulary.

To learn compositional programs, we follow Johnson et al. (2017a); Hu et al. (2017) and consider program generation as a sequence-to-sequence task. Different from prior approaches, our models are trained to fully generate the parameters of component modules in text. This approach is appropriate as reasoning programs in real data domains such as current video-grounded dialogues are usually shorter than those for synthetic data (Johnson et al., 2017a) and thus, program generation takes less computational cost. However, module parameters, i.e. entities and actions, contain much higher semantic variance than synthetic data, and our approach facilitates better transparency and interpretability. We adopt a simple template "$\langle \text{param}_1 \rangle \langle \text{module}_1 \rangle \langle \text{param}_2 \rangle \langle \text{module}_2 \rangle$..." as the target sequence. The resulting target sequences for dialogue and video understanding programs are sequences $\mathcal{P}_{\text{dial}}$ and $\mathcal{P}_{\text{vid}}$ respectively.

Table 1: Description of the modules and their functionalities. We denote $P$ as the parameter to instantiate each module, $H$ as the dialogue history, $Q$ as the question of the current dialogue turn, and $V$ as video input.

| Module | Input | Output | Description |
|---|---|---|---|
| find | P, H | $H_{ent}$ | For related entities in question, select the relevant tokens from dialogue history |
| summarize | $H_{ent}$, Q | $Q_{ctx}$ | Based on contextual entity representations, summarise the question semantics |
| where | P, V | $V_{ent}$ | Select the relevant spatial position corresponding to original (resolved) entities |
| when | P, $V_{ent}$ | $V_{ent+act}$ | Select the relevant entity-aware temporal steps corresponding to the action parameter |
| describe | P, $V_{ent+act}$ | $V_{ctx}$ | Select visual entity-action features based on non-binary question types |
| exist | Q, $V_{ent+act}$ | $V_{ctx}$ | Select visual entity-action features based on binary (yes/no) question types |

## 3.1 ENCODERS

**Text Encoder.** A text encoder is shared to encode text inputs, including dialogue history, questions, and captions. The text encoder converts each text sequence $\mathcal{X} = w_1, ..., w_m$ into a sequence of embeddings $X \in \mathbb{R}^{m \times d}$. We use a trainable embedding matrix to map token indices to vector representations of $d$ dimensions through a mapping function $\phi$. These vectors are then integrated with ordering information of tokens through a positional encoding function with layer normalization (Ba et al., 2016; Vaswani et al., 2017). The embedding and positional representations are combined through element-wise summation. The encoded dialogue history and question of the current turn are defined as $H = \text{Norm}(\phi(\mathcal{H}) + \text{PE}(\mathcal{H})) \in \mathbb{R}^{L_H \times d}$ and $Q = \text{Norm}(\phi(\mathcal{Q}) + \text{PE}(\mathcal{Q})) \in \mathbb{R}^{L_Q \times d}$.

To decode program and response sequences auto-repressively, a special token "*_sos*" is concatenated as the first token $w_0$. The decoded token $w_1$ is then appended to $w_0$ as input to decode $w_2$ and so on. Similarly to input source sequences, at decoding time step $j$, the input target sequence is encoded to obtain representations for dialogue understanding program $P_{\text{dial}}|_0^{j-1}$, video understanding program $P_{\text{vid}}|_0^{j-1}$, and system response $R|_0^{j-1}$. We combine vocabulary of input and output sequences and share the embedding matrix $E \in \mathbb{R}^{|\mathbb{V}| \times d}$ where $\mathbb{V} = \mathbb{V}^{in} \cap \mathbb{V}^{out}$.

**Video Encoder.** To encode video input, we use pre-trained models to extract visual features and audio features. We denote $F$ as the sampled video frames or video clips. For object-level visual features, we denote $O$ as the maximum number of objects considered in each frame. The resulting output from a pretrained object detection model is $Z_{\text{obj}} \in \mathbb{R}^{F \times O \times d_{\text{vis}}}$. We concatenate each object representation with the corresponding coordinates projected to $d_{\text{vis}}$ dimensions. We also make use of a CNN-based pre-trained model to obtain features of temporal dimension $Z_{\text{cnn}} \in \mathbb{R}^{F \times d_{\text{vis}}}$. The audio feature is obtained through a pretrained audio model, $Z_{\text{aud}} \in \mathbb{R}^{F \times d_{\text{aud}}}$. We passed all video features through a linear transformation layer with ReLU activation to the same embedding dimension $d$.

## 3.2 NEURAL MODULES

We introduce neural modules that are used to assemble an executable program constructed by the generated sequence from question parsers. We provide an overview of neural modules in Table 1 and demonstrate dialogue understanding and video understanding modules in Figure 3 and 4 respectively. Each module parameter, e.g. "a backpack", is extracted from the parsed program. For each parameter, we denote $P \in \mathbb{R}^d$ as the average pooling of component token embeddings.

**find(P,H)→$H_{ent}$**. This module handles entity tracing by obtaining a distribution over tokens in dialogue history. We use an entity-to-dialogue-history attention mechanism applied from an entity $P_i$ to all tokens in dialogue history. Any neural network that learn to generate attention between two tensors is applicable .e.g. (Bahdanau et al., 2015; Vaswani et al., 2017). The attention matrix normalized by softmax, $A_{\text{find,i}} \in \mathbb{R}^{L_H}$, is used to compute the weighted sum of dialogue history token representations. The output is combined with entity embedding $P_i$ to obtain contextual entity representation $H_{\text{ent,i}} \in \mathbb{R}^d$.

**summarize($H_{ent}$,Q)→$Q_{ctx}$**. For each contextual entity representation $H_{\text{ent,i}}$, $i = 1, ..., N_{\text{ent}}$, it is projected to $L_Q$ dimensions and is combined with question token embeddings through element-wise summation to obtain entity-aware question representation $Q_{\text{ent,i}} \in \mathbb{R}^{L_Q \times d}$. It is fed to a one-dimensional CNN with max pooling layer (Kim, 2014) to obtain a contextual entity-aware question representation. We denote the final output as $Q_{\text{ctx}} \in \mathbb{R}^{N_{\text{ent}} \times d}$.

While previous models usually focus on global or token-level dependencies (Hori et al., 2019; Le et al., 2019b) to encode question features, our modules compress fine-grained question representations

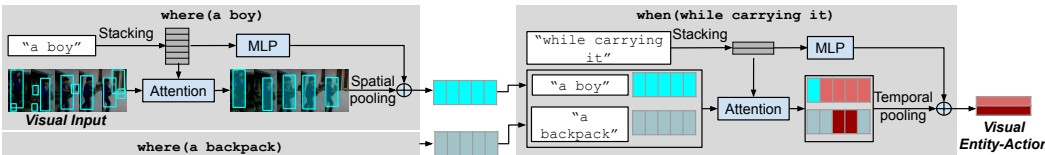

Figure 3: `find` and `summarize` neural modules for dialogue understanding

at entity level. Specifically, `find` and `summarize` modules can generate entity-dependent local and global representations of question semantics. We show that our modularized approach can achieve better performance and transparency than traditional approaches to encode dialogue context (Serban et al., 2016; Vaswani et al., 2017) (Section 4).

**where(P,V)**$\rightarrow$**V**$_{\text{ent}}$. Similar to the `find` module, this module handle entity-based attention to the video input. However, the entity representation $P$ in this case is parameterized by the original entity in dialogue rather than in question (See Section 3.3 for more description). Each entity $P_i$ is stacked to match the number of sampled video frames/clips $F$. An attention network is used to obtain entity-to-object attention matrix $A_{\text{where,i}} \in \mathbb{R}^{F \times O}$. The attended feature are compressed through weighted sum pooling along the spatial dimension, resulting in $V_{\text{ent,i}} \in \mathbb{R}^{F \times d}$, $i = 1, ..., N_{\text{ent}}$.

**when(P,V**$_{\text{ent}}$**)**$\rightarrow$**V**$_{\text{ent+act}}$. This module follows a similar architecture as the `where` module. However, the action parameter $P_i$ is stacked to match $N_{\text{ent}}$ dimensions. The attention matrix $A_{\text{when,i}} \in \mathbb{R}^F$ is then used to compute the visual entity-action representations through weighted sum along the temporal dimension. We denote the output for all actions $P_i$ as $V_{\text{ent+act}} \in \mathbb{R}^{N_{\text{ent}} \times N_{\text{act}} \times d}$

**describe(P,V**$_{\text{ent+act}}$**)**$\rightarrow$**V**$_{\text{ctx}}$. This module is a linear transformation to compute $V_{\text{ctx}} = W_{\text{desc}}^T[V_{\text{ent+act}}; P_{\text{stack}}] \in \mathbb{R}^{N_{\text{ent}} \times N_{\text{act}} \times d}$ where $W_{\text{desc}} \in \mathbb{R}^{2d \times d}$, $P_{\text{stack}}$ is the stacked representations of parameter embedding $P$ to $N_{\text{ent}} \times N_{\text{act}}$ dimensions, and $[;]$ is the concatenation operation.

The `exist` module is a special case of `describe` module where the parameter $P$ is the average pooled question embeddings. The above `where` module is applied to object-level features. For temporal-based features such as CNN-based and audio features, the same neural operation is applied along the temporal dimension. Each resulting entity-aware output is then incorporated to frame-level features through element-wise summation (Please refer to Appendix A.1).

Figure 4: `where` and `when` neural modules for video understanding

An advantage of our architecture is that it separates dialogue and video understanding. We adopt a transparent approach to solve linguistic entity references during the dialogue understanding phase. The resolved entities are fed to the video understanding phase to learn entity-action dynamics in video. We show that our approach is robust when dialogue evolves to many turns and video extends over time (Please see Section 4 and Appendix C).

### 3.3 DECODERS

**Question parsers.** The parsers decompose questions into sub-sequences to construct compositional reasoning programs for dialogue and video understanding. Each parser is an attention-based Transformer decoder. Given the encoded question $Q$, to decode program for dialogue understanding, the contextual signals are integrated through 2 attention layers: one attention on previously generated tokens, and the other on question tokens.

To generate programs for video understanding, the contextual signals are learned and incorporated in a similar manner. However, to exploit dialogue contextual cues, the execution output of dialogue

understanding neural modules $Q_{\text{ctx}}$ (See Section 3.2) is incorporated to each vector in $P_{\text{vid}}$ through an additional attention layer. This layer integrates the entity-dependent contextual representations from $Q_{\text{ctx}}$ to explicitly decode the original entities for video understanding programs.

**Response Decoder.** System response is decoded by incorporating the dialogue context and video context outputs from the corresponding reasoning programs to target token representations. We follows a vanilla Transformer decoder architecture (Le et al., 2019b), which consists of 3 attention layers: self-attention to attend on existing tokens, attention to $Q_{\text{ctx}}$ from dialogue understanding program execution, and attention to $V_{\text{ctx}}$ from video understanding program execution.

**Optimization**. We use the standard cross-entropy losses for prediction of dialogue and video understanding programs and output responses. We optimize models by joint training to minimize:

$$\mathcal{L} = \alpha\mathcal{L}_{\text{dial}} + \beta\mathcal{L}_{\text{vid}} + \mathcal{L}_{\text{res}} = \alpha\sum_j -\log(\mathbf{P}_{\text{dial}}(\mathcal{P}_{\text{dial,j}})) + \beta\sum_l -\log(\mathbf{P}_{\text{video}}(\mathcal{P}_{\text{video,l}})) + \sum_n -\log(\mathbf{P}_{\text{res}}(\mathcal{R}_{\text{n}}))$$

where $\mathbf{P}$ is the probability distribution of an output token. The probability is computed by passing output representations from the parsers and decoder to a linear layer $W \in \mathbb{R}^{d \times V}$ with softmax activation. We share the parameters between $W$ and embedding matrix $E$. The hyper-parameters $\alpha \geq 0$ and $\beta \geq$ are fine-tuned during training.

# 4 EXPERIMENTS

**Datasets.** We use the AVSD benchmark from the $7^{th}$ Dialogue System Technology Challenge (DSTC7) (Hori et al., 2019). In the experiments with AVSD, we consider two settings: one with video summary and one without video summary as input. In the setting with video summary, the summary is concatenated to the dialogue history before the first dialogue turn. We also adapt VilNMN to the video QA benchmark TGIF-QA (Jang et al., 2017). Different from AVSD, TGIF-QA contains a diverse set of tasks, which address different visual aspects in video.

**Training Procedure.** We follow prior approaches (Hu et al., 2017; 2018; Kottur et al., 2018) by obtaining the annotations of the programs through a language parser (Hu et al., 2016) and a reference resolution model (Clark & Manning, 2016). During training, we directly use these soft labels of programs and the given ground-truth responses to train the models. The labels are augmented with label smoothing technique (Szegedy et al., 2016). During inference time, we generate all programs and responses from given dialogues and videos. We run beam search to enumerate programs for dialogue and video understanding and dialogue responses. (Please see Appendix B for more details).

**AVSD Results.** We evaluate model performance by the objective metrics based on word overlapping, including BLEU (Papineni et al., 2002), METEOR (Banerjee & Lavie, 2005), ROUGE-L (Lin, 2004), and CIDEr (Vedantam et al., 2015), between each generated response and 6 reference gold responses. As seen in Table 2, our models outperform most of existing approaches. In particular, the performance of our model in the setting without video summary input is comparable to the GPT-based RLM (Li et al., 2020) with much smaller model size. The Student-Teacher baseline (Hori et al., 2019) specifically focuses on the performance gap between models with and without textual signals from video summary through a dual network of expert and student models. Instead, VilNMN reduces this performance gap through efficiently extracting relevant visual/audio information based on fine-grained entity and action signals. We also found that VilNMN applied on object-level features is competitive to the model applied on CNN-based features. The flexibility of VilNMN neural programs can also be seen in the experiment when the video understanding program is applied on the caption input as a visual feature.

**Ablation Analysis.** We experiment with several variants of VilMNM (either NMN or non-NMN-based) in the setting with CNN based features and video summary input As can be seen in Table 3, our approach to video and dialogue understanding through compositional reasoning programs exhibits better performance than non-compositional approaches. Compared to the approaches that directly process frame-level features in videos (Row B) or token-level features in dialogues (Row C, D), our full VilNMN (Row A) considers entity-level and action-level information extraction and thus, avoids unnecessary and possibly noisy extraction. Compared to the approaches that obtain dialogue contextual cues through a hierarchical encoding architecture (Row E, F) such as (Serban et al., 2016; Hori et al., 2019), VilNMN directly addresses the challenge of entity references in dialogues. As mentioned, we hypothesize that the hierarchical encoding architecture is more appropriate for less

Table 2: AVSD test results: The visual features are: I (I3D), ResNeXt-101 (RX), Faster-RCNN (FR), C (caption as a video input). The audio features are: VGGish (V), AclNet (A). ✓on PT denotes models using pretrained weights and/or additional finetuning. Best and second best results are bold and underlined respectively.

| Model | PT | Visual | Audio | BLEU4 | METEOR | ROUGE-L | CIDEr |
|---|---|---|---|---|---|---|---|
| **Audio/Visual only (without Video Summary/Caption)** | | | | | | | |
| Baseline (Hori et al., 2019) | - | I | - | 0.305 | 0.217 | 0.481 | 0.733 |
| Baseline (Hori et al., 2019) | - | I | V | 0.309 | 0.215 | 0.487 | 0.746 |
| Baseline+GRU+Attn. (Le et al., 2019a) | - | I | V | 0.315 | 0.239 | 0.509 | 0.848 |
| FGA (Schwartz et al., 2019) | - | I | V | - | - | - | 0.806 |
| JMAN (Chu et al., 2020) | - | I | - | 0.309 | 0.240 | 0.520 | 0.890 |
| Student-Teacher (Hori et al., 2019) | - | I | V | 0.371 | 0.248 | 0.527 | 0.966 |
| MTN (Le et al., 2019b) | - | I | - | 0.343 | 0.247 | 0.520 | 0.936 |
| MTN (Le et al., 2019b) | - | I | V | 0.368 | 0.259 | 0.537 | 0.964 |
| MSTN (Lee et al., 2020) | - | I | V | 0.379 | 0.261 | 0.548 | 1.028 |
| RLM-GPT2 (Li et al., 2020) | ✓ | I | V | **0.402** | 0.254 | 0.544 | 1.052 |
| VilNMN | - | I | - | 0.397 | 0.262 | **0.550** | **1.059** |
| VilNMN | - | FR | - | 0.388 | 0.259 | 0.549 | 1.040 |
| VilNMN | - | - | V | 0.381 | 0.252 | 0.534 | 1.004 |
| VilNMN | - | I | V | 0.396 | **0.263** | 0.549 | **1.059** |
| **Audio/Visual only (with Video Summary/Caption)** | | | | | | | |
| TopicEmb (Kumar et al., 2019) | - | I | A | 0.329 | 0.223 | 0.488 | 0.762 |
| Baseline+GRU+Attn. (Le et al., 2019a) | - | I | V | 0.310 | 0.242 | 0.515 | 0.856 |
| JMAN (Chu et al., 2020) | - | I | - | 0.334 | 0.239 | 0.533 | 0.941 |
| FA+HRED (Nguyen et al., 2018) | - | I | V | 0.360 | 0.249 | 0.544 | 0.997 |
| VideoSum (Sanabria et al., 2019) | - | RX | - | 0.394 | 0.267 | 0.563 | 1.094 |
| VideoSum+How2 (Sanabria et al., 2019) | ✓ | RX | - | 0.387 | 0.266 | 0.564 | 1.087 |
| MSTN (Lee et al., 2020) | - | I | V | 0.377 | 0.275 | 0.566 | 1.115 |
| Student-Teacher (Hori et al., 2019) | - | I | V | 0.405 | 0.273 | 0.566 | 1.118 |
| MTN (Le et al., 2019b) | - | I | - | 0.392 | 0.269 | 0.559 | 1.066 |
| MTN (Le et al., 2019b) | - | I | V | 0.410 | 0.274 | 0.569 | 1.129 |
| VGD-GPT2 (Le & Hoi, 2020) | ✓ | I | V | 0.436 | 0.282 | 0.579 | 1.194 |
| RLM-GPT2 (Li et al., 2020) | ✓ | I | V | **0.459** | **0.294** | **0.606** | **1.308** |
| VilNMN | - | I | - | 0.421 | 0.277 | 0.574 | 1.171 |
| VilNMN | - | FR | - | 0.421 | 0.275 | 0.571 | 1.148 |
| VilNMN | - | I | V | 0.421 | 0.277 | 0.573 | 1.167 |
| VilNMN | - | I+C | V | 0.429 | 0.278 | 0.578 | 1.188 |

Table 3: Ablation analysis of VilNMN with different model variants on the test split of the AVSD benchmark

| # | Model Variant | BLEU4 | CIDEr |
|---|---|---|---|
| A | Full VilNMN | **0.421** | **1.171** |
| B | ↪ No video NMNs; + vanilla text→video attention | 0.415 | 1.159 |
| C | ↪ No dial. NMNs; + response→history attention | 0.412 | 1.151 |
| D | ↪ No dial. NMNs; + response→concat(history+question) attention | 0.411 | 1.133 |
| E | ↪ No dial. NMNs; + $\text{HRED}_{\text{LSTM}}$(history) + question attn. | 0.414 | 1.153 |
| F | ↪ No dial. NMNs; + $\text{HRED}_{\text{GRU}}$(history) + question attn. | 0.415 | 1.138 |

entity-sensitive dialogues such as chit-chat and open-domain dialogues. Please see Appendix C for additional analysis of performance breakdown by dialogue turns and video lengths.

**Intepretability.** A difference of VilNMN from previous approaches to video-grounded dialogues is the model interpretability based on the predicted dialogue and video programs. From Figure 5, we observe that in cases where predicted dialogue programs and video program match or are close to the gold labels, the model can generate generally correct responses. In cases of wrong predicted responses, we can further look at how the model understands the questions based on predicted programs. In the $3^{rd}$ turn of example 1, the output response is missing a minor detail as compared to the label response because the video program fails to capture the parameter "rooftop". These subtle yet important details can determine whether output responses can fully address user queries. Similarly, in example 2, the model answers the question "what room" instead of question about "an object". For additional qualitative analysis, please see Appendix D.

**TGIF-QA Results.** In TGIF-QA experiments, we report the result using the L2 loss in *Count* task and accuracy in other tasks. From Table 4, VilNMN outperforms all baseline models in all tasks by a large margin. Compared to AVSD experiments, the TGIF-QA experiments emphasize video understanding ability of the models, removing the requirement for dialogue understanding and natural

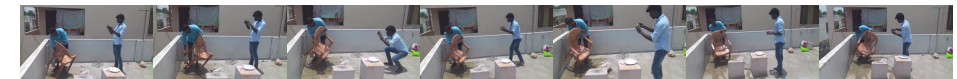

| Turn | Question | Dialogue NMN | Video NMN | Response |
|------|----------|--------------|-----------|----------|
| 3 | ...what are they doing in the scene ? are they on a rooftop ? | **Predicted**: `find(they), find(the scene) →summarize()` ✗ **Gold**: `find(they) →summarize()` | **Predicted**: `where(two men in the video),where(the scene)→ when(doing in the scene)→ describe(what)` ✗ **Gold**: `where(two men), where(rooftop) →when(doing in the scene) →describe(what)` | **Predicted**: one is washing a chair and the other is taking pictures ✗ **Gold**: yes , on a second floor roof deck , one man is washing a chair , another man is either filming or taking pictures |
| 4 | does he sit in the chair after washing it ? | **Predicted**: `find(he), find(the chair) →summarize()` ✓ **Gold**: `find(he), find(the chair)→ summarize()` | **Predicted**: `where(one man in the video), where(a chair)→ when(sit in the chair after washing it)→exist()` ✓ **Gold**: `where(one man), where(a chair)→ when(sit in the chair after washing it)→exist()` | **Predicted**: no , he does not sit in the chair ✓ **Gold**: no he does not , there is a pipe with water running all over |

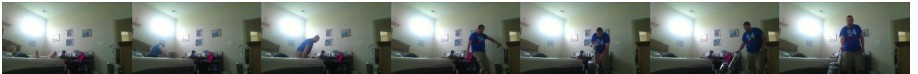

| Turn | Question | Dialogue Understanding Program | Video Understanding Program | Response |
|------|----------|-------------------------------|-----------------------------|----------|
| 3 | in the door way to the next room, there is an object. what is that? | **Predicted**: `find(the room), find(the door)→summarize()` ✗ **Gold**: `summarize()` | **Predicted**: `where(what room) →when(what is that)→exist()` ✗ **Gold**: `describe(what)` | **Predicted**: it looks like he is in a living room. ✗ **Gold**: he went to the doorway for a vacuum. |
| 4 | when he gets up, does he have anything in his hand? | **Predicted**: `find(he), find(his)→ summarize()` ✓ **Gold**: `find(he), find(his)→ summarize()` | **Predicted**: `where(one person in the video)→when(get up, have anything in his hands)→ describe(when)` ✗ **Gold**: `where(one person in the video)→when(get up), when(have anything in his hands)→ describe(when)` | **Predicted**: he has a vacuum in his hands. ✓ **Gold**: he goes for the vacuum. |

Figure 5: Intepretability of model outputs from a dialogue in the test split of the AVSD benchmark.

Table 4: Experiment results on the TGIF-QA benchmark. The visual features are: ResNet-152 (R), C3D (C), Flow CNN from two-stream model (F), VGG (V), ResNeXt-101 (RX).

| Model | Visual | Count (Loss) | Action (Acc) | Transition (Acc) | FrameQA (Acc) |
|-------|--------|--------------|--------------|------------------|---------------|
| VIS(avg) (Ren et al., 2015a) | R | 4.80 | 0.488 | 0.348 | 0.350 |
| MCB (aggr) (Fukui et al., 2016) | R | 5.17 | 0.589 | 0.243 | 0.257 |
| Yu et al. (Yu et al., 2017) | R | 5.13 | 0.561 | 0.640 | 0.396 |
| ST-VQA (t) (Gao et al., 2018) | R+F | 4.32 | 0.629 | 0.694 | 0.495 |
| Co-Mem (Gao et al., 2018) | R+F | 4.10 | 0.682 | 0.743 | 0.515 |
| PSAC (Li et al., 2019) | R | 4.27 | 0.704 | 0.769 | 0.557 |
| HME (Fan et al., 2019) | R+C | 4.02 | 0.739 | 0.778 | 0.538 |
| STA (Gao et al., 2019) | R | 4.25 | 0.723 | 0.790 | 0.566 |
| CRN+MAC (Le et al., 2019c) | R | 4.23 | 0.713 | 0.787 | 0.592 |
| MQL (Lei et al., 2020) | V | - | - | - | 0.598 |
| QueST (Jiang et al., 2020) | R | 4.19 | 0.759 | 0.810 | 0.597 |
| HGA (Jiang & Han, 2020) | R+C | 4.09 | 0.754 | 0.810 | 0.551 |
| GCN (Huang et al., 2020) | R+C | 3.95 | 0.743 | 0.811 | 0.563 |
| HCRN (Le et al., 2020) | R+RX | 3.82 | 0.750 | 0.814 | 0.559 |
| VilNMN | R | **2.65** | **0.845** | **0.887** | **0.747** |
| ↪ soft label programs | R | 1.90 | 0.857 | 0.898 | 0.780 |
| ↪ - res-to-question attn. | R | 3.28 | 0.801 | 0.776 | 0.679 |

language generation. This is demonstrated through higher performance gaps between VilNMN with generated programs and soft label programs as compared to ones in AVSD experiments. We also observe that an attention layer attending to question is important during the response decoding phase in TGIF-QA as there is no dialogue context $Q_{\text{ctx}}$ in Video QA tasks.

## 5 CONCLUSION

While conventional neural network approaches have achieved notable successes in video-grounded dialogues and video QA, they often rely on superficial pattern learning principles between contextual cues from questions/dialogues and videos. In this work, we introduce Visio-Linguistic Neural Module Network (VilNMN). VilNMN consists of dialogue and video understanding neural modules, each of which performs entity and action-level operations on language and video components. Our comprehensive experiments on AVSD and TGIF-QA benchmarks show that our models can achieve competitive performance while promoting a compositional and interpretable learning approach.

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

# A    ADDITIONAL MODEL DETAILS

## A.1    NEURAL MODULES ON TEMPORAL FEATURES

To adapt our neural modules to temporal features, we apply the same neural architectures in all modules except for the `where` module. In object-level features, this module operates on object-based or spatial-based level. We can apply this module to temporal-based features similarly simply by not stacking the parameter and pooling the attended features along the temporal dimension. For an entity parameter $P_i$, the attention matrix in this case is an entity-to-temporal-step matrix $A_{\text{where,i}} \in \mathbb{R}^F$ and the resulting pooled feature is $V_{\text{ent,i}} \in \mathbb{R}^d$. Before feeding this representation to a `when` module, we incorporate each $V_{\text{ent,i}}$ into feature of each temporal step through an MLP layer and element-wise summation, resulting in $V_{\text{ent,i}}^{stack} \in \mathbb{R}^{\text{F} \times d}$ where $F$ is the number of sampled video frames/clips. An overview of the `where` module with temporal features can be seen in Figure 6.

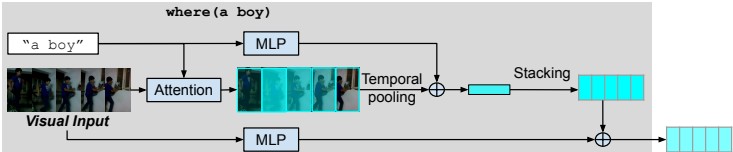

Figure 6: Adaptation of the `where` module to temporal-based features

We adapt this module in a similar manner to other temporal-level features such as audio and textual features such as video caption. We keep the same architecture in the `when` module. We denote the resulting output from the `when` module for all actions $P_i$ is $V_{\text{act}} \in \mathbb{R}^{N_{\text{act}} \times d}$. We concatenate this to the output from the previous `where` module $V_{\text{ent}}$ to obtain $V_{\text{ent+act}} \in \mathbb{R}^{(N_{\text{ent}} + N_{\text{act}}) \times d}$. This is used as input to the `describe` or `exist` module.

## A.2    QUESTION PARSER

The parsers decompose questions into sub-sequences to construct compositional reasoning programs for dialogue and video understanding. Each parser is an attention-based Transformer decoder. The Transformer attention is a multi-head attention on query, key, and value tensors, denoted as $\text{Attention}(\text{Query}, \text{Key}, \text{Value})$. For each token in the Query sequence , the distribution over tokens in the Key sequence is used to obtain the weighted sum of the corresponding representations in the Value sequence.

$$\text{Attention}(Query, Key, Value) = \text{softmax}(\frac{QueryKey^T}{\sqrt{d_{key}}})Value \in \mathbb{R}^{L_{\text{query}} \times d_{\text{query}}}$$

Each attention is followed by a feed-forward network applied on each position identically. We exploit the multi-head and feed-forward architecture, which show good performance in NLP tasks such as NMT and QA (Vaswani et al., 2017; Dehghani et al., 2019), to efficiently incorporate contextual cues from dialogue components to parse question into reasoning programs. Given the encoded question $Q$, to decode program for dialogue understanding, the contextual signals are integrated through 2 attention layers: one attention on previously generated tokens, and the other on question tokens. At time step $j$, we denote the output from an attention layer as $A_{\text{dial,j}}$.

$$A_{\text{dial}}^{(1)} = \text{Attention}(P_{\text{dial}}|_0^{j-1}, P_{\text{dial}}|_0^{j-1}, P_{\text{dial}}|_0^{j-1}) \in \mathbb{R}^{j \times d}$$
$$A_{\text{dial}}^{(2)} = \text{Attention}(A_{\text{dial}}^{(1)}, Q, Q) \in \mathbb{R}^{j \times d}$$

Similarly, to generate programs for video understanding, the contextual signals are learned and incorporated in a similar manner. However, to exploit dialogue contextual cues, the execution output of dialogue understanding neural modules $Q_{\text{ctx}}$ is incorporated to each vector in $P_{\text{dial}}$ through an additional attention layer. This layer integrates the resolved entity information to decode the original entities for video understanding. It is equivalent to a reasoning process that converts the question from its original multi-turn semantics to single-turn semantics.

$$A_{\text{vid}}^{(3)} = \text{Attention}(A_{\text{vid}}^{(2)}, Q_{\text{ctx}}, Q_{\text{ctx}}) \in \mathbb{R}^{j \times d}$$

## A.3 NON-NMN MODELS

For ablation analysis, we evaluate several variants of VilNMN, based on the following categories:

To test the contribution of our NMN approach for video understanding, we remove the parser for video understanding program and related neural modules and replace them with pure neural network architecture (*Model B*). Specifically, we remove neural modules `where`, `when`, `describe`, and `exist`. We then directly use video feature embeddings $V$ as $V_{\text{ctx}}$ as input to the original attention layer in response decoder similarly to (Hori et al., 2019; Sanabria et al., 2019).

$$A_{\text{res}}^{(3)} = \text{Attention}(A_{\text{res}}^{(2)}, V, V) \in \mathbb{R}^{j \times d}$$

To further test the contribution of NMN architecture for dialogue understanding, we similarly remove the question parser for dialogue understanding program and neural modules `find` and `describe`. We then directly use the dialogue history embeddings $H$ and question embeddings $Q$ as inputs to the response decoder in two different ways. First, we replace the original attention on dialogue context $Q_{\text{ctx}}$ with two attention layers to attend on dialogue history and question sequentially (*Model C*). As noted by Le et al. (2019b), question input contains much more relevant signals than dialogue history and attention operation should be separated from the one on dialogue history.

$$A_{\text{res}}^{(2a)} = \text{Attention}(A_{\text{res}}^{(1)}, H, H) \in \mathbb{R}^{j \times d}$$
$$A_{\text{res}}^{(2b)} = \text{Attention}(A_{\text{res}}^{(2a)}, Q, Q) \in \mathbb{R}^{j \times d}$$

Alternatively, we simply concatenate dialogue and question embeddings similarly to (Hori et al., 2019; Sanabria et al., 2019) and use it as input to the original attention layer (*Model D*).

$$A_{\text{res}}^{(2)} = \text{Attention}(A_{\text{res}}^{(1)}, [H; Q], [H; Q]) \in \mathbb{R}^{j \times d}$$

To use more sophisticated neural models for dialogue understanding, we further adopt the hierarchical encoding architecture with question attention (Li et al., 2016; Serban et al., 2016; Hori et al., 2019). Each dialogue turn $\mathcal{H}_t$, including a pair of human utterance and system response, is processed separately by a word-level RNN such as LSTM (*Model E*) or GRU (*Model F*). A sentence-level RNN is used to sequentially process the last hidden states obtained previously turn by turn. The output in each recurrent step is fed to an attention layer such as (Bahdanau et al., 2015; Vaswani et al., 2017) to obtain question-aware representations of dialogue history.

$$H_t^{\text{word}} = \text{RNN}(H_t) \in \mathbb{R}^d$$
$$H_t^{\text{sent}} = \text{RNN}(H_t^{\text{word}}) \in \mathbb{R}^d$$
$$H = [H_t^{\text{sent}}]|_{t=1}^{T-1} \in \mathbb{R}^{d \times (T-1)}$$
$$Q_{\text{ctx}} = \text{Attention}(Q, H, H) \in \mathbb{R}^{L_Q \times d}$$

where $T$ is the current dialogue turn. The output is treated as $Q_{\text{ctx}}$ and is fed to the corresponding attention layer in the response decoder.

## B ADDITIONAL EXPERIMENT DETAILS

### B.1 DATASETS

We use the AVSD benchmark from DSTC7 (Hori et al., 2019) which consists of dialogues grounded on the Charades videos (Sigurdsson et al., 2016). Each dialogue contains up to 10 dialogue turns, each turn consists of a question and expected response about a given video. For visual features, we use the 3D CNN based features from a pretrained I3D model (Carreira & Zisserman, 2017) and object-level features from a pretrained FasterRNN model (Ren et al., 2015b). The audio features are obtained from a pretrained VGGish model (Hershey et al., 2017). In the experiments with AVSD, we consider two settings: one with video summary and one without video summary as input. In the setting with video summary, the summary is concatenated to the dialogue history before the first dialogue turn. We also adapt VilNMN to the video QA benchmark TGIF-QA (Jang et al., 2017). Different from AVSD, TGIF-QA contains a diverse set of QA tasks:

- *Count*: open-ended task which counts the number of repetitions of an action

- *Action*: multiple-choice (MC) task which asks about a certain action occurring for a fixed number of times
- *Transition*: MC task which emphasizes temporal transition in video
- *Frame*: open-ended task which can be answered from visual contents of one of video frames

For the TGIF-QA benchmark, we use the extracted features from a pretrained ResNet model (He et al., 2016).

Table 5: Summary of DSTC7 AVSD and TGIF-QA benchmark

|  | # | Train | Val. | Test |
|---|---|---|---|---|
| **AVSD** | Dialogs | 7,659 | 1,787 | 1,710 |
|  | Turns | 153,180 | 35,740 | 13,490 |
|  | Words | 1,450,754 | 339,006 | 110,252 |
| **TGIFQA** | Count QA | 24,159 | 2,684 | 3,554 |
|  | Action QA | 18,428 | 2,047 | 2,274 |
|  | Trans. QA | 47,434 | 5,270 | 6,232 |
|  | Frame QA | 35,453 | 3,939 | 13,691 |

## B.2 TRAINING DETAILS

We use a training batch size of 32 and embedding dimension $d = 128$ in all experiments. Where Transformer attention is used, we fix the number of attention heads to 8 in all attention layers. In neural modules with MLP layers, the MLP network is fixed to 2 linear layers with a ReLU activation in between. In neural modules with CNN, we adopt a vanilla CNN architecture for text classification (without the last MLP layer) where the number of input channels is 1, the kernel sizes are $\{3, 4, 5\}$, and the number of output channels is $d$. We initialize models with uniform distribution (Glorot & Bengio, 2010). During training, we adopt the Adam optimizer (Kingma & Ba, 2015) and a decaying learning rate Vaswani et al. (2017) where we fix the warm-up steps to 15K training steps. We employ dropout (Srivastava et al., 2014) of 0.2 at all networks except the last linear layers of question parsers and response decoder. We train models up to 50 epochs and select the best models based on the average loss per epoch in the validation set.

## C ADDITIONAL RESULTS

To evaluate model robustness, we report the relative performance by calculating the difference of CIDEr in experimental settings against the most basic setting. Specifically, we compare against performance of output responses in the first dialogue turn position (i.e. $2^{nd}$-$10^{th}$ turn vs. the $1^{st}$ turn), or responses grounded on the shortest video length range (video ranges are intervals of 0-$10^{th}$, 10-$20^{th}$ percentile and so on). We report the results of the model variants A, B, and E (See the Ablation Analysis section in the main paper and Appendix A.3 for model description). First, as can be seen in Figure 7, for various dialogue turn positions, we observe that the original VilNMN (model A) suffers less than model E when dialogues extend over time up the $8^{th}$ turn. This explains the contribution of dialogue understanding modules in solving entities even when the dialogues grow longer. Secondly, as compared to model B, we observe that the Full VilNMN (model A) is less affected as the videos grounding the dialogues grow longer. The difference is clear when the video length increases up to 33 seconds.

We also report the absolute scores and compare model variants. In Table 6a, we compare model variants B and E. We observe that model B generally performs better than model E in overall, especially in higher turn positions, i.e. from the $4^{th}$ turn to $8^{th}$ turn. Interestingly, we note some mixed results in very low turn position, i.e. the $2^{nd}$ and $3^{rd}$ turn, and very high turn position, i.e. the $10^{th}$ turn. Potentially, in very high turn position, the neural based approach such as hierarchical RNN can better capture the global dependencies within dialogue context than the entity-based compositional NMN method.

In Table 6b, we compare model variants A and B. We note that the performance gap between model A and B is quite distinct, with 7/10 cases of video ranges in which model A outperforms. However, similarly to our prior observations in experiments by dialogue turn, in lower ranges (i.e. 1-23

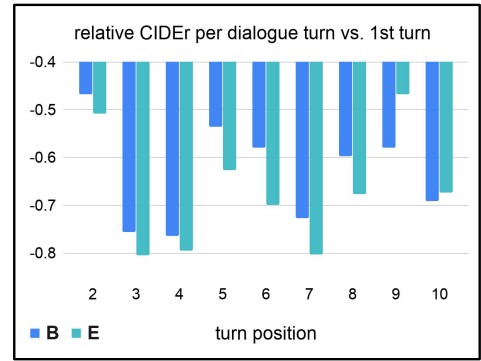 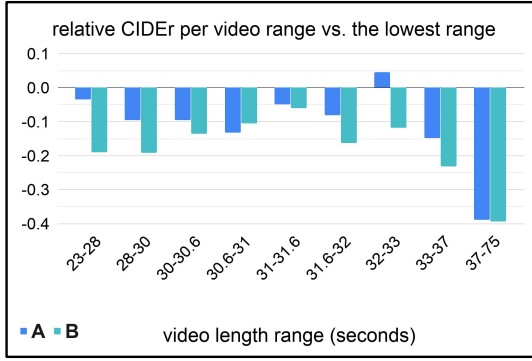

Figure 7: Performance of model variants A, B, and E, by dialogue turn position and video length. The performance is calculated relatively to performance of the most basic setting, i.e. responses of the first dialogue turn $\Delta\text{CIDEr}_{\text{turn\_i}} = \text{CIDEr}_{\text{turn\_i}} - \text{CIDEr}_{\text{turn\_1}}$, or responses grounding on the lowest video range (0 to 23 seconds) $\Delta\text{CIDEr}_{\text{range\_i}} = \text{CIDEr}_{\text{range\_i}} - \text{CIDEr}_{0-23}$.

Table 6: Performance breakdown in BLEU4 and CIDEr

(a) by dialogue turn between model variants B and E.

| | BLEU4 | | CIDEr | |
|---|---|---|---|---|
| turn position | Model B | Model E | Model B | Model E |
| 1 | 0.579 | **0.587** | 1.623 | **1.650** |
| 2 | 0.429 | **0.430** | **1.155** | 1.142 |
| 3 | 0.275 | **0.289** | **0.867** | 0.846 |
| 4 | **0.309** | 0.305 | **0.859** | 0.855 |
| 5 | **0.355** | 0.335 | **1.088** | 1.023 |
| 6 | **0.357** | 0.329 | **1.044** | 0.950 |
| 7 | **0.342** | 0.325 | **0.896** | 0.847 |
| 8 | **0.361** | 0.332 | **1.025** | 0.973 |
| 9 | 0.383 | **0.431** | 1.043 | **1.182** |
| 10 | **0.395** | 0.371 | 0.931 | **0.977** |

(b) by video length range (in seconds) between model variants A and B.

| | BLEU4 | | CIDEr | |
|---|---|---|---|---|
| video range (seconds) | Model A | Model B | Model A | Model B |
| 1-23 | 0.432 | **0.447** | 1.298 | **1.355** |
| 23-28 | **0.436** | 0.433 | **1.264** | 1.165 |
| 28-30 | **0.398** | 0.376 | **1.203** | 1.164 |
| 30-30.6 | **0.441** | 0.418 | **1.220** | 1.202 |
| 30.6-31 | **0.413** | 0.411 | **1.250** | 1.166 |
| 31-31.6 | 0.439 | **0.451** | 1.249 | **1.295** |
| 31.6-32 | **0.430** | 0.419 | **1.217** | 1.192 |
| 32-33 | **0.468** | 0.445 | **1.343** | 1.237 |
| 33-37 | **0.388** | 0.381 | **1.149** | 1.124 |
| 37-75 | 0.356 | **0.365** | 0.910 | **0.962** |

seconds) and higher ranges (37-75 seconds), model A performs not as well as model B. There are additional factors that we will need to examine further to explain the results, such as the complexity of the questions for these short and long-range videos. Potentially, our question parser for video understanding program needs more sophisticated composition method to retrieve information from these video ranges.

# D    QUALITATIVE ANALYSIS

We extract the predicted programs and responses for some example dialogues in Figure 8, 9, 10, and 11 and report our observations:

- We observe that when the predicted programs are correct, the output responses generally match the ground-truth (See the $1^{st}$ and $2^{nd}$ turn in Figure 8, and the $1^{st}$ and $4^{th}$ turn in Figure 10) or close to the ground-truth responses ($1^{st}$ turn in Figure 9).

- When the output responses do not match the ground truth, we can understand the model mistakes by interpreting the predicted programs. For example, in the $3^{rd}$ turn in Figure 8, the output response describes a room because the predicted video program focuses on the entity "what room" instead of the entity "an object" in the question. Another example is the $3^{rd}$ turn in Figure 10 where the entity "rooftop" is missing in the video program. These mismatches can deviate the information retrieved from the video during video program execution, leading to wrong output responses with wrong visual contents.

- We also note that in some cases, one or both of the predicted programs are incorrect, but the predicted responses still match the ground-truth responses. This might be explained as the predicted module parameters are not exactly the same as the ground truth but they are close enough (e.g. $4^{th}$ turn in Figure 8). Sometimes, our model predicted programs that are more appropriate than the ground truth. For example, in the $2^{nd}$ turn in Figure 9, the program is added with a `where` module parameterized by the entity "the shopping bag" which was solved from the reference "them" mentioned in the question.

- We observe that for complex questions that involve more than one queries (e.g. the $3^{rd}$ turn in Figure 10), it becomes more challenging to decode an appropriate video understanding program and generate responses that can address all queries.

- In Figure 11, we demonstrate some output examples of VilNMN and compare with two baselines: Baseline (Hori et al., 2019) and MTN (Le et al., 2019b). We noted that VilNMN can include important entities relevant to the current dialogue turn to construct output responses while other models might miss some entity details, e.g. "them/dishes" in example A and "the magazine" in example B. These small yet important details can determine the correctness of dialogue responses.

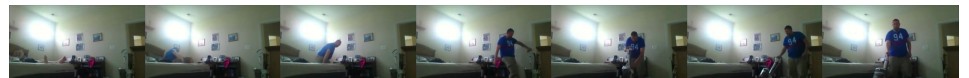

| Turn | Question | Dialogue Understanding Program | Video Understanding Program | Response |
|---|---|---|---|---|
| 1 | how many people are in the video ? | **Predicted**: `summarize()` ✓ 
 **Gold**: `summarize()` | **Predicted**: `describe(how many people)` ✓ 
 **Gold**: `describe(how many people)` | **Predicted**: there is only one person in the video. ✓ 
 **Gold**: there is one person in the video. |
| 2 | does this person ever leave the field of view ? | **Predicted**: `find(this person)→summarize()` ✓ 
 **Gold**: `find(this person)→ summarize()` | **Predicted**: `where(one person in the video)→when(leave the field of view)→exist()` ✓ 
 **Gold**: `where(one person in the video)→when(leave the field of view)→exist()` | **Predicted**: no, he does not leave the view. ✓ 
 **Gold**: no, he is in the video the whole time. |
| 3 | in the door way to the next room, there is an object. what is that? | **Predicted**: `find(the room), find(the door)→summarize()` ✗ 
 **Gold**: `summarize()` | **Predicted**: `where(what room)→when(what is that)→exist()` ✗ 
 **Gold**: `describe(what)` | **Predicted**: it looks like he is in a living room. ✗ 
 **Gold**: he went to the doorway for a vacuum. |
| 4 | when he gets up, does he have anything in his hand? | **Predicted**: `find(he), find(his)→summarize()` ✓ 
 **Gold**:`find(he), find(his)→summarize()` | **Predicted**: `where(one person in the video)→when(get up, have anything in his hands)→ describe(when)` ✗ 
 **Gold**: `where(one person in the video)→when(get up), when(have anything in his hands)→ describe(when)` | **Predicted**: he has a vacuum in his hands. ✓ 
 **Gold**: he goes for the vacuum. |

Figure 8: Example output programs and responses from a dialogue in the test split of the AVSD benchmark.

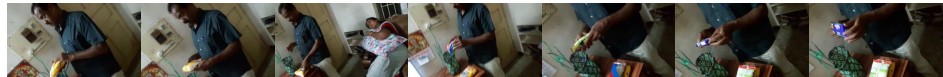

| Turn | Question | Dialogue Understanding Program | Video Understanding Program | Response |
|---|---|---|---|---|
| 1 | what is is holding in his hand in the beginning ? | **Predicted**: `find(his)→summarize()` ✓ 
 **Gold**: `find(his)→summarize()` | **Predicted**: `where(a man)→when(holding in his hand in the beginning)→ describe(what)` ✓ 
 **Gold**: `where(a man)→ when(holding in his hand in the beginning)→ describe(what)` | **Predicted**: he is holding a shopping bag in his hand . ✗ 
 **Gold**: looks like a bag of beans |
| 2 | does he put them down ? | **Predicted**: `find(he), find(them)→summarize()` ✗ 
 **Gold**: `find(he)→summarize()` | **Predicted**: `where(a man), where(a shopping bag)→when(put them down)→exist()` ✗ 
 **Gold**: `where(a man)→ when(put them down)→exist()` | **Predicted**: yes he puts them down on the table . ✓ 
 **Gold**: yes he puts it down after looking at it |

Figure 9: Example output programs and responses from a dialogue in the test split of the AVSD benchmark.

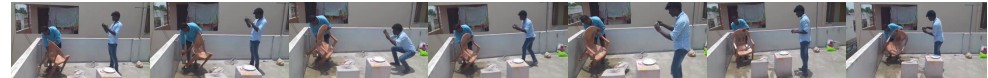

| Turn | Question | Dialogue Understanding Program | Video Understanding Program | Response |
|---|---|---|---|---|
| 1 | hi how many people are in the video from start to finish ? | **Predicted**: `summarize()` ✓
**Gold**: `summarize()` | **Predicted**: `describe(how many people)` ✓
**Gold**: `describe(how many people)` | **Predicted**: there are two people in the video . ✓
**Gold**: hello , there are two men |
| 2 | are they carrying on a conversation between *themselves* ? | **Predicted**: `find(they) →summarize()` ✓
**Gold**: `find(they), find(themselves) →summarize()` | **Predicted**: `where(two men in the video)→when(carrying on a conversation between themselves)→exist()` ✓
**Gold**: `where(two men)→ when (carrying a conversation between themselves)→exist()` | **Predicted**: yes , they are carrying a conversation ✗
**Gold**: no , background noise is birdsong and shuffling noises |
| 3 | what are they doing in the scene ? are they on a rooftop ? | **Predicted**: `find(they), find(the scene) →summarize()` ✗
**Gold**: `find(they) →summarize()` | **Predicted**: `where(two men in the video),where(the scene)→ when(doing in the scene)→ describe(what)` ✗
**Gold**: `where(two men), where(rooftop)→when(doing in the scene) →describe(what)` | **Predicted**: one is washing a chair and the other is taking pictures ✗
**Gold**: yes , on a second floor roof deck , one man is washing a chair , another man is either filming or taking pictures |
| 4 | does he sit in *the chair* after washing it ? | **Predicted**: `find(he), find(the chair) →summarize()` ✓
**Gold**: `find(he), find(the chair)→ summarize()` | **Predicted**: `where(one man in the video), where(a chair)→ when(sit in the chair after washing it)→exist()` ✓
**Gold**: `where(one man), where(a chair)→ when(sit in the chair after washing it) →exist()` | **Predicted**: no , he does not sit in the chair ✓
**Gold**: no he does not , there is a pipe with water running all over |

Figure 10: Example output programs and responses from a dialogue in the test split of the AVSD benchmark.

| Video (Sample Frame) | Dialog History and Question | Predicted Dialogue & Video Understanding Program | Predicted and Ground-truth Responses |
|---|---|---|---|
| **A** | **Dialog History**: ...what is going on when the video starts ? a man is looking out a doorway...what is he doing in there then ? he starts to wash dishes ...
**Question**: does he wipe *them* with a towel? | **Dialog Program (VilNMN)**: `find(he), find(them) →summarize()`
**Video Program (VilNMN)**: `where(a man), where(the dishes) → when(wipe them with a towel) → exist()` | **Predicted (VilNMN)**: no he doesn't wipe them with a towel . ✓
**Predicted (Baseline)**: no he does not take a picture of the camera. ✗
**Predicted (MTN)**: no he does not do anything with with hands. ✗
**Gold**: looks like he just uses his hands . |
| **B** | **Dialog History**: ...what is he reading? He is reading a magazine and he is laughing...
**Question**: is he laughing because of *the magazine*? | **Dialogue Program (VilNMN)**: `find(he), find(the magazine) → summarize()`
**Dialogue Program (VilNMN)**: `where(one young boy in the video), where(a magazine) → when (laughing because of the magazine)→ exist()` | **Predicted (VilNMN)**: yes, he is laughing at the magazine . ✓
**Predicted (Baseline)**: yes he is talking to the camera . ✗
**Predicted (MTN)**: no he is not laughing at the end of the video . ✗
**Gold**: maybe , because then he throws the magazine aside . |
| **C** | **Dialog History**: ...how many people are in the video? There is one person in the video...
**Question**: in the door *way* to the *next room*, there is *an object*. What is that? | **Dialogue Program (VilNMN)**: `find(the room), find(the door)→ summarize()`
**Dialogue Program (VilNMN)**: `where(what room) → when (what is that)→ exist()` | **Predicted (VilNMN)**: it looks like he is in a living room . ✗
**Predicted (Baseline)**: i m not sure what it is . ✗
**Predicted (MTN)**: he walks into the room . ✗
**Gold**: he went to the doorway for a vacuum . |

Figure 11: Intepretability of example outputs from VilNMN and baselines models (Hori et al., 2019; Le et al., 2019b)

