# OpenReview forum: "VilNMN: A Neural Module Network approach to Video-Grounded Language Tasks"
_ICLR.cc/2021/Conference — Reject_

### Official Review · AnonReviewer2 · 2020-10-26
**I am glad to see the progress in applying NMN in video-grounded language tasks but the paper could be improved on its current form.**

**Rating:** 5
**Confidence:** 5

**Review:**

Summary:

The paper studies the application of neural module network to video-grounded language tasks. They propose a method dubbed Visio-Linguistic Neural Module Network (VilNMN) to retrieve spatio-temporal information in a video through a linguistic-based parsed program. In particular, VilNMN first extracts entity references and their corresponding actions in linguistic cues. This information is then being used to locate relevant information in the visual cue to arrive at the correct answer. The proposed method is evaluated on two large scales benchmarks AVSD and TGIF-QA, demonstrating competitive performance with state-of-the-art methods.

Comments (Technical, Major Flaws of this paper):

(1) Overall, it is interesting to see neural module network works in such a complex setting as in video-grounded language tasks and I appreciate the efforts of the authors trying to explain the method as detailed as possible.

(2) The use of the mathematical symbols in the paper is very confusing. Normally, upper cases (capital letters) are used to denote matrices/sets of vectors while lower cases (non-capital letters) are often used to denote vectors.

(3) How did the authors subsample F frames/clips from a video? In my understanding, there are a lot of frames in a video are blurry or distorted so if you sample them in a random manner, it would greatly affect the performance of object detection (Faster RCNN in this case). Please elaborate more on this.

(4) Subsection 3.1, in the description of "when" module: A_when,i is a vector, not a matrix. Same in the supplementary document. In addition, I am wondering since there are many objects/entities in a video, how would the "when" module be able to localize the same object/entity over time given no additional supervision? I think object tracking would greatly be beneficial in this case.

(5) In the experiments on TGIF-QA:
- Why the authors only used ResNet features instead of using similar features as in AVSD? Spatial-based level features are fine in terms of computations but are less intuitive as I expect the object-based features counterpart and the linguistic entity references represent things at the same level of abstraction.
- Honestly, I am skeptical about the results on the TGIF-QA datasets as the gap between VilNMN and the existing methods is very significant. From Table 4, it looks like motion features (optical flow, C3D, and the likes) play a role in tasks containing repetition of actions (such as count and action). I am not sure the reason why VilNMN without the use of any of those motion features could manage to outperform the existing methods with large margins?
- Please provide more analysis on the results on the TGIF-QA as in my understanding even when a model correctly links entities in a question with their visual representations, it does not guarantee that it can arrive at correct answers. For example, in counting task, I would say most of the questions have the same parsed program. How does your model work in this scenario?


Some other concerns:

(6) At the end of the related works, the authors wrote "In video represented as sequence of images,...., e.g though average pooling, resulting in potential loss of information". This is a big assumption as using average pooling over object proposals is a bad idea as object appearance may vary very little over time in a video and the average pooling would smash the temporal information in the video. If one can properly model object tubelets via object tracking, what written in the paper wouldn't make sense.

(7) Section 3.2: The sentence "...calculated as softmax scores between an entity P_i and each token in dialogue history" is mathematically incorrect. What drives the attention weights? Softmax is a function applying over a set of elements.

I would be okay to raise the rating if the authors sufficiently address my concerns during the rebuttal phase.

---

> ### Author Response · Authors · 2020-11-24
> **Response to Reviewer 2**
>
> We thank R2 for the detailed feedback. We appreciate that the reviewer found this work an interesting method using NMN in complex video-grounded language tasks. Please find below our response:
>
> * Concern #1:
> “how would the "when" module be able to localize the same object/entity over time given no additional supervision? I think object tracking would greatly be beneficial in this case.”
>
> In our current approach, we pass the outputs from the ‘where’ modules applied on all entities/objects into the next “when” module. One motivation for this is that most of the question utterances in dialogues are usually a single sentence length. Each sentence often contains multiple subjects/objects but less than one action-related phrase. As suggested by the reviewer, object tracking might improve the reasoning performance of the reasoning steps between ‘when’ and ‘where’ modules in this case. We will try to investigate this direction in the future.
>
> * Concern #2:
>  “Why the authors only used ResNet features instead of using similar features as in AVSD?”
>
> Thank you for your suggestion! We did not have enough time to report the results using an object-based feature counterpart. We reported the performance using ResNet-152 to fairly compare with the baseline models. For completeness, we will try to report similar results using features such as F-RCNN in the final version.
>
> * Concern #3:
> “More analysis on the results on the TGIF-QA”
>
> About the results on the TGIF-QA benchmark, our performance gain can be explained as it comes from the linguistic bias decoded as reasoning structure programs. The decoded programs allow more precise neural (attention) operations between detected entities/actions and the video features. Specifically, in TGIF-QA, the questions are designed to follow a fixed question type distribution with certain question templates. This makes it much easier for the question parser to parse program structures. In our experiments, using exact-match accuracy of parsed programs vs. label programs to gauge performance of the question parser, our question parser can achieve a performance 81% to 94% accuracy in TGIF-QA vs. 41-45% in AVSD.
>
>
> * Concern #4:
> “...using average pooling over object proposals is a bad idea as object appearance may vary very little over time in a video and the average pooling would smash the temporal information in the video.”
>
> Thank you for your feedback. We added a discussion on the use of object tracking in this part in the related work (Section 2). Compared to a solution using an object tracking mechanism, our method adopts a multi-step interaction framework between the space-time information in video with entity-action detected in text.
>
> * Concern #5:
> Other questions:
>
> 1. “How did the authors subsample F frames/clips from a video?”
>
> We sampled F frames/clips with a striding of 8 frames from a video.
>
> 2. Typos and presentation issues:
>
> Thank you for your feedback! We fixed some of the errors and will thoroughly check again in the final version.

---

### Official Review · AnonReviewer1 · 2020-10-27
**Adapting Neural Module Network (NMN) for Video dialog and QA**

**Rating:** 5
**Confidence:** 3

**Review:**

Description:

This paper introduces the Visio-Linguistic Neural Module Network (VilNMN) consisting of a pipeline of dialogue and video understanding neural modules. Motivated by Hu et al. (2017), Kottur et al (2017), this paper extends the NMNs on video tasks for interpretable neural models. The model explicitly resolves entity references (dialog understanding) and detects actions from videos (video understanding) for response generation. Experiments show that NMNs achieve competitive results on AVSD (video-dialog) and TGIF-QA (video-QA) benchmarks.

Strengths:
- New modules for video understanding (“where”, “when”) have been proposed. A step towards interpretability of compositional neural networks
- Ablation studies have been provided to understand the importance of each module in the VILNMN model.
- The breakdown of relative CIDEr/BLEU (Figure 7 in supplementary) for different context and video length is interesting.
- SOTA results on TGIF-QA (Video QA) while competitive results on AVSD (dialog task).

Weaknesses:
- Availability of code is not discussed which is essential for reproducibility.
- It would help to specify whether sentence vs corpus level BLEU was used for evaluation
- Human evaluation is not provided. Limitations of automatic metrics and their reliability in language generation have been discussed repeatedly. See (Reiter and Belz, 2009; Novikova et al., 2017; Reiter, 2018).
- Table 6 denotes that the evaluation results decrease with longer video or larger context modeling which is the main focus of the paper.
- It might be argued that this approach would not generalize. How would this model scale when the dialog becomes challenging (in terms of disfluencies, ellipses or alignment, topic switch, etc apart from co-reference; see Haber et al 2019 Photobook dataset for brief summary of dialog phenomena)? Similarly when the videos become more complex, would action recognition suffice?
- CorefNMN (Kottur et al 2017) was designed specifically for co-reference resolution in the dialog. The paper could be similarly improved by explicitly motivating the specific utility of the NMN modules compared to high-level description- eg. to capture co-reference (dialog) and action recognition (video).
- An analysis of the AVSD dataset would help understand the importance of the dialog context in the dataset - focus of one of the modules in this paper. (See Agarwal et al 2020 study for Visual dialog)

Questions:
- Since AVSD is also posed as a retrieval task (Alamri et al. 2019), have the authors evaluated the system on ranking based metrics?
- Could the authors clarify why accuracy is also not reported for CountQA in Table 4?
- It would help to explicitly mention the neural modules previously defined and the novel modules, eg how the “find” module differs from Kottur et al. 2018.
- Have the authors experimented with the dialog-based modules from Kottur et al. 2018 - eg. “refer” module?
- In Fig 2 (as well as the main text), it would help to clarify if the underlying text encoder is shared for the dialog history, question, and caption.
- How are the audio signals incorporated in the VilNMN mentioned in Table 2? Pardon if I missed this.
- Have the authors experimented with pre-trained weights (decoders)?

Suggestions/Comments:
- Previously Johnson et al. (2017a); Hu et al. (2017), Kottur et al. (2018) have all explored NMN for visio-linguistic tasks (such as VQA, Visual dialog), the nomenclature “Visio-Linguistic Neural Module Network (VLNMN)” seems too broad. Something on the grounds of “ActionNMN/ActNMN” would do justice to the work.
- Model descriptions of ablations in Table 3 could be improved for clarity.
- Implementation details could be further specified - the framework, all other hyperparameters to ease reproducibility.

--------------------------------------------------------------------------------------------------------------------------------------------------------
Post Rebuttal update:

I would like to thank the authors for answering the questions. I believe that an updated version addressing all the concerns in detail will find its place in other future conferences. Original rating is maintained.

---

> ### Author Response · Authors · 2020-11-24
> **Response to Reviewer 1**
>
> We thank R1 for the valuable feedback. We appreciate that the reviewer found this work “a step towards interpretability of compositional neural networks.” Please find below our response:
>
> * Concern #1:
> “It might be argued that this approach would not generalize.”
>
> Thank you for your suggestion on the potential dialogue phenomena. In this work, we constraint our models to dialogues and/or question-answering problems with videos as the common grounding information. We design our reasoning structures following simple thought processes such as ‘find’, ‘when’, and ‘where’ to extract information about entities and actions. We think that the simplicity of the method makes it easier to generalize to different scenarios of dialogues and videos. We hope to extend this method in the future and investigate specific challenges in dialogues and/or videos in other scenarios.
>
> * Concern #2:
> “The specific utility of the NMN modules compared to high-level description.”
>
> In Table 1, we included an overview of the NMN modules with their intended functionality in our NMN model pipeline. The high-level description of modules are to address information retrieval in time-space dimensions through the corresponding 'when' and 'where' module.
>
> * Concern #3:
> “Human evaluation is not provided”
>
> We noted that human evaluation is more reliable than automatic metrics in language generation tasks. However, in the AVSD benchmark, it has shown that the human score is closely correlated with automatic metric, as shown in the Dialogue Technology Challenge 7 (https://arxiv.org/abs/1901.03461). This is due to the simpler problem setting in the benchmark, closely similar to question-answering problems.
>
> * Concern #4:
> “Availability of code is not discussed which is essential for reproducibility.”
>
> We will make the code available for reproducibility, including details of training and hyper-parameters.
>
> *Concern #5:
> Other questions:
>
> 1. “It would help to specify whether sentence vs corpus level BLEU was used for evaluation”:
>
> We follow the evaluation used in the previous baseline models. The BLEU score is computed by sentence level, between generated responses and 6-reference ground truth.
>
> 2. “Could the authors clarify why accuracy is also not reported for CountQA in Table 4?”
>
> We follow the evaluation method from previous baseline models and report the loss value, computed as the mean squared error between outputs and labels.
>
> 3. “How are the audio signals incorporated in the VilNMN mentioned in Table 2?”
>
> The audio signals are incorporated using the same method with some modification to make it compatible to temporal-only features. We included further details in Appendix A.
>
> 4. “An analysis of the AVSD dataset would help understand the importance of the dialog context in the dataset - focus of one of the modules in this paper”
>
> Thank you for the suggestion! We tried to use the ablation results to explain the importance of dialogue context. Specifically, in Appendix C, we reported the results of generated responses at different turn positions (hence, different sizes of dialogue context).
>
> 5. “Since AVSD is also posed as a retrieval task (Alamri et al. 2019), have the authors evaluated the system on ranking based metrics?”
>
> Thank you for your suggestion. We will try to experiment on this setting and report in the final version. As we noted, the retrieval task dataset is currently only available in train and validation splits (https://video-dialog.com/).
>
> 6. “Have the authors experimented with pre-trained weights (decoders)?”
>
> Other than using models to extract visual features, we did not experiment with pre-trained weights on decoders. It is an interesting direction and we will try to include the results in the final version.
>
> 7. Writing and presentation issues:
>
> Thank you for all the feedback! We have improved the manuscripts based on some of your comments and will thoroughly check again in the final version, including considering changing our method name to another term.

---

### Official Review · AnonReviewer3 · 2020-10-29
**This paper present a novel NMN approach to solving video-grounding lanuguage tasks, which decompses all language into entity references and detect corresponding action-based visual feature, then instantiate NMN with those inputs to get the final response.**

**Rating:** 4
**Confidence:** 4

**Review:**

The author present a novel neural moudalar network  for video grounding tasks, which can provide interpretable intermediate reasoing outcomes and show the model robustness.
This model achieves competitive results on AVSD datasets and state-of-the-art performance on TGIF-QA datasets, which demonstrates the effectiveness of the model design.

Detailed comments are listed in the following
• The novelity of the NMN is limited in this paper. The similari idea have been used in many previous literatures. I am wondering that how you define the modular space? Is there any prinpicle guidelines to design module like "find, summarize, when, describe"?
• The reasoning struture in this papar is simple. The module "find, when, where" are more like signal detectors. There is no reasoning structure for how to get the final response. (in this paper, just fuse the detected information to get the final answer by a response decoder). So this methods cannot reveal the inner correlation between final response and detected visual/language entities.
• [Question] How do you train the program generation tasks from language? Is there any groundtruth program structure annotation to supervise this? How do you determine the hyper-parameters \alpha and \beta?
• The paper is written pooly and some expressions are confusing, like "Different from..., our model are trained to fully genrate the parameters of components in the text". The parameter here refer the input of each module, which is different from model parameters.

---

> ### Author Response · Authors · 2020-11-24
> **Response to Reviewer 3**
>
> We thank R3 for the detailed feedback. We are glad that the reviewer found our approach “a novel neural modular network for video grounding tasks, which can provide interpretable intermediate reasoning outcomes and show the model robustness.” Please find below our response:
>
> * Concern #1:
> “I am wondering how you define the modular space?”
>
> We design our modular space on the principle of time-space/spatio-temporal dimension in video. The spatial information in video can be defined as information regarding entities, which inspires us for a ‘where’ neural module. The temporal information in video contains information regarding changes from one frame to another. This is a key difference from traditional neural modular network models and we extend the prior approaches with a ‘when’ module. While the reasoning structure is simple, it is easier to adapt to different problem scenarios, especially in real world domains such as the AVSD benchmark. Our experiments show that predicted reasoning structures can provide additional bias to generate better dialogue responses. In our qualitative results, we demonstrated both failure and success examples that output responses can be explained by the predicted reasoning structures (Figure 5 and Appendix D).
>
> * Concern #2:
> “How do you train the program generation tasks from language?”
>
> We train the program generation tasks through the question parsers. We obtained supervision labels for this task using available linguistic parsers. We then train the parsers as a sequence-to-sequence task, with source sequences as the text inputs (dialogue context, question of the current turn) and the target sequence is the program. Please refer to Section 3.3. and Appendix A for more details.
>
> * Concern #3:
> “How do you determine the hyper-parameters \alpha and \beta?”
>
> The hyper-parameters are fine-tuned during training and we selected the best combination using the model response generation performance on the validation split.

---

### Official Review · AnonReviewer4 · 2020-10-30
**Interesting idea; results need further clarification**

**Rating:** 5
**Confidence:** 2

**Review:**

This paper studies the language grounding aspect of video-language problems. It proposes a Neural Module Network (NMN) for explicit reasoning of visually-grounded object/action entities and their relationships. The proposed method is demonstrated to be somewhat effective in the audio-visual dialogue task and has been shown superior to existing works on video QA. Overall, the paper is motivated clearly and is delivered with good clarity. The followings need to be clarified.

i) The proposed model demonstrates impressive results on TGIF-QA but without any insightful justification. Since the questions in TGIF-QA are short and usually do not involve complicated reasoning, intuitively, a heavy reasoning scheme might not necessarily pay off. Please clarify the performance gain and possible reasons. Also, "soft label programs" lack the necessary context (and should be in bold instead in Tab. 4).

ii) Including intense model variants in the main result table (Tab. 2) gives this paper a somewhat unfair advantage, especially when the best performing method on each metric comes from different model variants. The validation set (from both AVSD and TGIF-QA) is supposed to serve the purpose of model architecture search and ablation studies. Besides, the underlines in the lower part of Tab. 2 should go to method VGD-GPT2.

========== Post-Rebuttal ==========

Concerns on paper/results clarity still persist. Lowering my rating to 5.

---

> ### Author Response · Authors · 2020-11-24
> **Response to Reviewer 4**
>
> We thank the R4 for all the feedback. We are glad that R4 mentioned “the paper is motivated clearly and is delivered with good clarity.” Please find below our response:
>
> * Concern #1:
>  “The proposed model demonstrates impressive results on TGIF-QA but without any insightful justification”:
>
> Even though the questions in TGIF-QA are short, they still contain a compositional structure (i.e. entities, actions) that previous approaches are not explicitly trained to learn. In our work, the models are trained to detect the compositionality in questions explicitly. Since TGIF-QA questions follow a very specific question type distribution (count, action, transition, and frameQA), the question structures are simpler and easier to learn than AVSD. Using exact-match accuracy of parsed programs vs. label programs as a metric, our question parser can achieve a performance 81% to 94% accuracy in TGIF-QA vs. 41-45% in AVSD. The higher accuracy in decoding a reasoning structure leads to better performance in predicting the output answers.
>
> * Concern #2:
> “Including intense model variants in the main result table (Tab. 2) gives this paper a somewhat unfair advantage, especially when the best performing method on each metric comes from different model variants.”
>
> The model variants in Table 2 are differentiated by the data settings but the architecture is the same. In the AVSD benchmark, multiple types of data/modalities are considered (visual, audio) and we presented the performance based on different combinations of modalities. For a comparison, we also presented all baseline model performance based on their data settings as well.
>
> * Concern #3:
> Typos and presentation issues:
>
> Thank you for your feedback! We fixed some of the errors and will thoroughly check again in the final version.

---

### Decision · Program_Chairs · 2021-01-07
**Final Decision**

**Decision:**

Reject

**Comment:**

The authors propose a neural module based approach for reasoning about video grounding.  The goal is to provide performance and interpretability.  Unfortunately, the reviewers found the paper opaque, the results confusing, and expressed repeated concerns about the novelty, fairness of comparisons and concerns that the surprising results were not sufficiently well justified by the paper (or the author's response).